**Data Availability Statement:** All relevant data are within the paper and its Supporting information files.

# Association of childhood trauma, and resilience, with quality of life in patients seeking treatment at a psychiatry outpatient: A cross-sectional study from Nepal

Saraswati Dhungana[1,2]*, Rishav Koirala[2,3‡], Saroj Prasad Ojha[1‡], Suraj Bahadur Thapa[1,2,4]

1 Department of Psychiatry and Mental Health, Institute of Medicine, Tribhuvan University, Kathmandu, Nepal, 2 Division of Mental Health and Addiction, Institute of Clinical Medicine, University of Oslo, Oslo, Norway, 3 Brain and Neuroscience Center, Kathmandu, Nepal, 4 Division of Mental Health and Addiction, Oslo University Hospital, Oslo, Norway

These authors contributed equally to this work.
‡ RK and SPO also contributed equally to this work.
* saraswati.dhungana@mmc.tu.edu.np, saraswati.dhungana@studmed.uio.no, iomsaras@gmail.com

## Abstract

Quality of life is defined by the World Health Organization as "Individuals' perception of their position in life in the context of the culture and value systems in which they live and in relation to their goals, expectations, standards and concerns". It is a comprehensive measure of health outcome after trauma. Childhood maltreatment is a determinant of poor mental health and quality of life. Resilience, however, is supposed to be protective. Our aim is to examine childhood trauma and resilience in patients visiting psychiatry outpatient and investigate their relations with quality of life. A descriptive cross-sectional study was conducted with a hundred patients with trauma and visiting psychiatry outpatient. Standardized tools were applied to explore childhood trauma, resilience, quality of life and clinical diagnoses and trauma categorization. Sociodemographic and relevant clinical information were obtained with a structured proforma. Bivariate followed by multivariate logistic regressions were conducted to explore the relation between childhood trauma, resilience, and quality of life. Poor quality of life was reported in almost one third of the patients. Upper socioeconomic status, emotional neglect during childhood, current depression and low resilience were the determinants of poor quality of life in bivariate analysis. Final models revealed that emotional neglect during childhood and low resilience had independent associations with poor quality of life. Efforts should be made to minimize childhood maltreatment in general; and explore strategies to build resilience suited to the cultural context to improve quality of life.

## Introduction

Childhood maltreatment has been defined "as any act or series of acts of commission or omission by a parent or other caregiver that results in harm, potential for harm, or threat of harm

**Funding:** SBT received the funding from The Norwegian Partnership Programme for Global Academic Cooperation (NORPART). NORPART grant 2018/10039 from Norwegian Agency for International Cooperation and Quality Enhancement in Higher Education (DIKU)". However, the funding agency was not involved in research designing, data collection and analysis, planning and/ or manuscript preparation or decision to publish.

**Competing interests:** The authors have declared that no competing interests exist.

to a child" [1]. This has been studied extensively in the last few decades [2, 3]. Studies have revealed that childhood maltreatment is not an uncommon occurrence with most children exposed to at least one form of maltreatment [4–6]. Reported prevalence rates of childhood maltreatment in high income countries vary from one country to another, ranging from 5–6% for both sexes in Norway to 27% girls and 30% in boys for physical abuse in the United States. Similarly, the rates of sexual abuse reported was 10–14% in girls and 3–4% in boys in the Norwegian general population, while it was higher for boys up to 6% and for girls as high as 14% in a meta-analysis conducted in Europe in 2011, with even higher rates in the US, 25% in girls and 16% in boys [5, 7–10]. Studies on prevalence and forms of maltreatment in the low- and middle-income countries context are scant with rates even higher [11]. A study from India among 12–18 years adolescents with history of child workforce reported rates above 80% for any form of maltreatment, while physical abuse was the most common form reported in almost two-thirds of the participants [12]. Studies from Nepal report similar estimates with some variations on the predominant form of trauma [13–15].

Childhood maltreatment has long-term adverse consequences on health through a number of pathways, extending from biological networks to psychological vulnerability [16]. Most studies point towards a consistent, and negative correlation between childhood trauma and a number of adverse physical and mental health conditions, including quality of life [7, 13, 17–19], adding on to the societal burden and economic costs [7, 20, 21]. Cumulative traumatic events in context of childhood trauma have been linked to further adversities later in life [22] and impaired quality of life [18, 23].

On the other hand, resilience has been found to have a protective effect on later mental health outcomes [24]. In face of the highly prevalent nature of trauma in general population globally, resilience is key in maintaining an optimal health outcome [25]. Resilience is, therefore, a key component in trauma research [25] although there are discussions on the complex and multisystemic nature of resilience [24, 26]. Individuals who are resilient are said to have better mental health profiles [24, 27], including better quality of life.

Quality of life (QOL) is defined by the World Health Organization as "Individuals' perception of their position in life in the context of the culture and value systems in which they live and in relation to their goals, expectations, standards and concerns". It therefore, is a multidimensional construct of health that goes beyond clinical diagnoses. Studies suggest that there is relationship between resilience and quality of life [27] but the nature of the relation between the two has not been studied extensively, especially in context of trauma and childhood trauma.

Nepal is a lower- middle income country in South Asia, with 17.4% of the population still under poverty line as of 2021. Studies suggest that children who have been exposed to poverty and adverse social circumstances are more at risk of experiencing childhood traumatic events [3, 28, 29]. Furthermore, physical acts of discipline are common practices in children and young people embedded within the culture with similar rates of emotional and other traumatic events [13, 30] in Nepali population. To our knowledge, scant literature is available from Nepal [13–15, 30] on childhood maltreatment prevalence and correlates but no studies exploring relation with QOL.

The current study thereby attempts to fulfill this gap by exploring the prevalent types of childhood trauma individuals report when they come to seek treatment at a psychiatric outpatient of a university hospital in Kathmandu, Nepal. We also investigated the relationship between most kinds of childhood trauma, resilience, and quality of life in these individuals. We hypothesized that QOL is poor in those with positive childhood trauma history and has positive relation with resilience.

## Methods

### Participants

This study had a cross sectional descriptive design and was a part of a broader study "Study of health outcomes after trauma" (SHOT) [31, 32]. The participants in this study were adults 18–60 years presenting to psychiatry outpatient of Tribhuvan University Teaching hospital, a tertiary hospital in Kathmandu, Nepal. The eligibility criteria were at least one trauma exposure before one month prior to the outpatient visit, operationalized as defined in the ICD-10 PTSD section K [33]. Those with dyslexia, cerebral infection, severe head injury, serious medical or neurological illness, organic mental disorders, and psychotic disorders were excluded from the study. We used purposive sampling technique for the study participants. Sample size was calculated based on the Cochran's formula of sample size calculation for cross- sectional studies as follows:

$$n = (z1 - \alpha/2)^{2*}(p)(q)/(d)^2, n = (z1 - \alpha/2)^{2*}(p)(q)/(d)^2.$$

Here, p = 50% from previous study [34], $(z_{1-} \alpha_{/2})^2$ = 1.96 for 95% confidence interval, and d = 10%, precision of estimate. With this, the sample size calculated, n = 96.04 ~ 96. Considering 10% dropouts, we calculated the total required sample size as 96+10 = 106. We, however, were able to recruit only 100 patients in our study. However, for a few participants, observations on some variables were missing. We had missing information on childhood trauma for two patients and on resilience and QOL measure for one patient so we have used the data from 98 of the participants for childhood trauma measure and 99 for QOL and resilience measure.

### Procedure

Patients fulfilling the inclusion criteria for the study and who consented to participate were provided detailed information regarding the project in verbal and in written form. They were also informed about their voluntariness in willing to withdraw from the study at any time during data collection without the need for justification.

Written informed consent was then obtained from all participants deemed eligible for the study. Informants accompanying the participants provided consent for those not able to read and / or write. Ethical approval for this study was obtained from the Nepal Health Research Council (NHRC) (reference number 801) and Institutional Review Committee (IRC) at the Institute of Medicine (reference number 480 (611)6$^2$/ 075/076) in Nepal and Research Ethical Committee (REK) in Norway (reference number 2015/2081). Details about the methodology are published elsewhere [32, 35].

A predesigned proforma was used to gather sociodemographic and other relevant information on trauma. We categorized gender as binary, age into two groups as less or equal to 24 and more than 24 years, marital status into two as single and married, religion as Hindu and others, residence as rural and urban, education as illiterate and literate and socioeconomic status into two as lower and upper. Trauma variables of interest were trauma numbers into two as single and repetitive and time since trauma elapsed into three as trauma less than 1 year, 1–10 years and more than 10 years.

### Study measures

**WHO Quality of Life-Brief version (WHOQOL-BREF).** The WHOQOL-BREF is a shorter 26- item version of the original WHOQOL 100-item scale. It is a comprehensive measure of QOL incorporating multiple domains, with good psychometric properties. The total

scores are computed by summing up the scores on each domain as specified in the manual [36]. The Nepali translation by Giri et al. [37] was used in this study, where we categorized the total QOL score into two categories as good (total QOL score more than 45) and poor (total QOL score less or equal to 45) [38].

**Child Trauma Questionnaire Short Form (CTQ-SF).** The Childhood Trauma Questionnaire-Short Form (CTQ-SF) [39] is the gold standard in regard to assessing childhood traumas and encompasses five aspects, namely emotional abuse, physical abuse, emotional neglect, physical neglect, and sexual abuse. It is brief, convenient with good psychometric properties across clinical samples as a retrospective measure of childhood maltreatment. For our purpose, we used the Nepali translated and adapted version by Kohrt et al. [40, 41]. This version excluded the statements referring to sexual abuse for cultural adaptation purposes. In Nepali culture, questions regarding sexual abuse are considered inappropriate and offensive even to this date, making the participants uncomfortable. The cut-off score used for both physical neglect and physical abuse was nine, whereas the cut-off for emotional abuse was twelve and emotional neglect was fourteen as reported in the original.

**Wagnild and Young Resilience scale (RS).** The Wagnild and Young Resilience scale (RS) is a shorter eight item version of the original 25- item scale [42]. The original scale comprises two domains, namely personal competence with seventeen items and acceptance of self and life domain with eight items, respectively. For our study, we used the Nepali translated and adapted version [41]. This is a Likert type scale, where higher mean scores point towards better resilience.

**WHO World Mental Health Composite International Diagnostic Interview (WMH-CIDI) version 2.1.** WHO-CIDI 2.1 is a standardized diagnostic interview guide developed by WHO, which corresponds to psychiatric diagnoses in ICD-10 and DSM-IV. The psychometric properties are reported to be good in studies [33]. We used the Nepali translation with standard guidelines [43]. We used sections D, E and K for diagnoses of generalized anxiety disorder, major depressive disorder, and post-traumatic stress disorder, respectively. We also used the list of ten traumatic events listed in section K to categorize trauma.

## Statistics

Stata 17 was used for all statistical purposes (Stata Corp LLC, College Station, TX, USA). Normal distribution of continuous variables was assessed using histogram and boxplots. Then after, means, medians, standard deviations and inter quartile ranges were applied for continuous variables and frequencies with percent for categorical variables for descriptive statistics. For inferential statistics, chi-square tests were applied for bivariate association between categorical variables followed by multivariate logistic regression in the final models. Variables found significant in the bivariate analyses, along with other relevant variables from literature such as age, gender, and current psychiatric diagnoses, were first checked for confounding and those with variation inflation factor less than 2 were entered in the final models to test our hypotheses. Hosmer and Lemeshow test were done to check for model fit. P value .05 was considered significant for all statistical purposes.

## Results

There were 99 participants in this study. Almost one-third of the participants had a poor quality of life. Upper socioeconomic status and emotional neglect type of childhood trauma along with low resilience and current major depression had significant associations with poor QOL.

## Sociodemographic profile of participants and QOL

Table 1 shows the sociodemographic correlates of participants and QOL.

## Childhood trauma types, time since trauma, other trauma and QOL

Table 2 depicts the association of childhood trauma types, time since trauma, number of traumas and QOL.

## Current psychiatric disorders and QOL

Table 3 shows the association of psychiatric disorders, and QOL.

## Factors independently associated with QOL

Table 4 shows all the factors associated with QOL. Bivariate analyses were first done followed by multivariate logistic regressions to check if childhood traumatic events and resilience had statistically significant associations in predicting QOL. Variables that were shown to be statistically significant in the bivariate models were entered into the final models along with other relevant variables such as age, gender, and other current psychiatric diagnoses from published literature. After adjustment in the final models, only childhood trauma type emotional neglect,

**Table 1. Sociodemographic variables and quality of life.**

| Variables ($n$ = 99) | Quality of life | | $\alpha^2$ | CI (LB, UB) |
|---|---|---|---|---|
| | Poor(33) | Better(66) | | |
| Gender | | | | |
| Male 48 | 16 | 32 | -0.0 | .43, 2.33 |
| Female 51 | 17 | 34 | | |
| Age (years) | | | | |
| Less or equal to 24 | 5 | 14 | 0.52 | .22, 2.03 |
| More than 24 | 28 | 52 | | |
| Marital status | | | | |
| Single | 5 | 15 | 0.78 | .20, 1.27 |
| Married | 28 | 51 | | |
| Religion | | | | |
| Hindu | 28 | 53 | 0.30 | .44, 2.44 |
| Others | 5 | 13 | | |
| Residence | | | | |
| Rural | 19 | 30 | 1.30 | .70, 0.38 |
| Urban | 14 | 36 | | |
| Education | | | | |
| Illiterate | 7 | 10 | 0.55 | 0.52, 4.40 |
| Literate | 26 | 56 | | |
| SES | | | | |
| Lower | 14 | 44 | 5.32* | .16, 0.87 |
| Upper | 19 | 22 | | |

$\alpha^2$ = chi- square statistic, CI = Confidence interval, LB = Lower bound, UB = Upper bound, SES = socioeconomic status

*p<0.05

**Table 2. Child trauma, trauma variables and quality of life.**

| Variables (*n* = 98) | QOL | | $\alpha^2$ | CI (LB, UB) |
|---|---|---|---|---|
| | Poor | Better | | |
| Trauma number | | | | |
| Single | 15 | 26 | 0.33 | 0.55, 2.98 |
| Repetitive | 18 | 40 | | |
| Time since trauma in years | | | | |
| Less than 1 | 5 | 9 | 0.51 | |
| 1–10 | 23 | 50 | | .36, 4.00 |
| More than 10 | 5 | 7 | | .16, 3.8 |
| CTQ | | | | |
| CTQ Physical abuse | | | | |
| No | 23 | 46 | 0.05 | 0.44, 2.82 |
| Yes | 9 | 20 | | |
| CTQ Emotional abuse | | | | |
| No | 24 | 56 | 1.34 | 0.19, 1.52 |
| Yes | 8 | 10 | | |
| CTQ Physical neglect | | | | |
| No | 13 | 28 | 0.03 | 0.39, 2.19 |
| Yes | 19 | 38 | | |
| CTQ Emotional neglect | | | | |
| No | 22 | 58 | 4.97* | 0.11, 0.87 |
| Yes | 10 | 8 | | |

n = number of participants, $\alpha^2$ = chi-square statistic, CI = Confidence interval LB = lower bound, UB = upper bound, CTQ = child trauma questionnaire, QOL = quality of life,
*p<0.05

resilience scores and current depression were found to have significant associations with better QOL.

## Discussion

This study explores QOL in context of childhood trauma and resilience taken into consideration in the context of Nepal. The findings from this study extend support to the existing literature suggesting poor quality of life in patients with childhood trauma. Almost one-third of patients who sought treatment in psychiatry outpatient department reported poor quality of life. Though literature on exact prevalence on QOL estimates is scant, studies report impaired QOL in survivors of major trauma [44]. Many factors contribute to poor QOL in such studies.

Our study concluded that QOL was better in those with lower socioeconomic status in bivariate analyses. This is in striking contrast to most studies reporting a positive correlation between the two [45, 46]. SES is a robust measure of health comprising three most essential elements, namely education, employment, and family income [47]. It is the single most important social determinant of overall health and quality of life. However, the association is complex and is a function of a number of biological, psychological, social, and cultural factors [48]. Exploring other confounders might have given a more accurate picture. The possibility of having more depression/ emotional neglect/ childhood trauma or other relevant factors such as social support, in those with higher SES is also a valid concern prone to give biased estimates in terms of association with QOL. Also, the nature of the measure both in terms of SES and the

**Table 3. Psychiatric disorders and QOL.**

| Variables (n = 98) | QOL | | $\alpha^2$ | CI (LB, UB) |
|---|---|---|---|---|
| | Poor | Better | | |
| PTSD | | | | |
| No | 27 | 56 | 0.15 | 0.26, 2.44 |
| Yes | 6 | 10 | | |
| Depression | | | | |
| No | 11 | 56 | 26.44*** | 0.03,0.24 |
| Yes | 22 | 10 | | |
| GAD | | | | |
| No | 17 | 44 | 2.11 | 0.23, 1.25 |
| Yes | 16 | 22 | | |
| PTSD+depression | | | | |
| No | 29 | 60 | 0.22 | 0.19, 2.77 |
| Yes | 4 | 6 | | |
| PTSD+GAD | | | | |
| No | 31 | 61 | 0.08 | 0.23, 6.92 |
| Yes | 2 | 5 | | |

n = number of participants, $\alpha^2$ = chi-square statistic, CI = Confidence interval, LB = Lower bound, UB = Upper bound, PTSD = Post traumatic stress disorder,

GAD = Generalized anxiety disorder, PTSD = post-traumatic stress disorder, QOL = quality of life,

***$p < 0.001$

QOL and the context of their usage need to be considered before making generalization. The rest of the sociodemographic variables did not show statistical significance.

It is interesting to note that those with history of emotional neglect form of trauma in childhood had independent association with poor quality of life in our study, with other traumas having no effect. This was a surprising result since several studies examining the relationship between various childhood trauma types and adult QOL, including systemic reviews [18, 49] have reported otherwise. Childhood trauma is related to poor QOL regardless of the type of

**Table 4. Factors associated with better QOL.**

| Variables (n = 98) | Bivariate analysis (Unadjusted model) | Multivariate analysis (Adjusted model) |
|---|---|---|
| | OR (CI) | OR (CI) |
| Age (≤24 year) | 0.66 (0.22, 2.03) | .17 (0.02, 1.25) |
| Gender (Male) | 1 (0.43, 2.31) | 0.96 (0.25, 3.72) |
| SES (Lower) | 0.37 (0.16, 0.87)* | 0.61 (0.16, 2.34) |
| CTQ Emotional neglect (No) | 0.30 (0.11, -0.87)* | 0.07 (0.01, 0.43)** |
| Current depression (No) | 0.09 (0.03, 0.24)*** | 0.05 (0.01, 0.26)*** |
| Current PTSD (No) | 0.80 (0.26, 2.44) | 2.62 (0.46, 14.80) |
| Current GAD (No) | 0.53 (0.23, 1.25) | 0.87 (0.21, 3.55) |
| Mean resilience score | (2.13, 6.44)*** | 4.52 (1.97, 10.40)*** |

n = number of participants, OR = odds ratio, CI = Confidence interval, CTQ = child trauma questionnaire,

PTSD = post-traumatic stress disorder, GAD = generalized anxiety disorder, SES = socio-economic status,

*$p < 0.05$

**$p < 0.01$

***$p < 0.001$

trauma. This was also one of our hypotheses on starting to write the paper, but during analysis, we found that only emotional neglect was associated with QOL, unlike in most studies. Psychological maltreatment, a broader term encompasses both abuse and neglect and has been shown to be associated with an equal and even greater number of behavioral problems and disorders [50]. Studies have consistently demonstrated a clear relationship between most kinds of childhood trauma [16, 18, 51], ranging from physical abuse, physical neglect, emotional neglect, emotional abuse, and sexual abuse [2, 16]. Neglect of any kind is as harmful as physical and sexual abuse though has received less attention [8]. QOL assessment during childhood and adolescence also demonstrated clear negative association with any kind of childhood maltreatment [52, 53]. It is important to note that these effects extend beyond adulthood [54]. This could be for the fact that there is heterogeneity in terms of the measure of childhood trauma and childhood maltreatment, as much as the study design and study population. One of the most influential papers in childhood maltreatment and its trajectory later in life studied multiple forms of maltreatment, where many family problems such as mental illness in family, mothers exposed to violence were also given same scores as other forms such as neglect or abuse [4]. Biased estimates in terms of maltreatment reporting are also possible depending on whether it is self-reported or parental report [3]. Ours is a study conducted in those seeking treatment at psychiatry facility and there was also history of another trauma considered besides the childhood trauma variable. As much as the presence of childhood trauma, we also know that the perception of trauma, the nature of trauma on whether it is a recurrent event, social circumstances of trauma and more importantly, social support following trauma are more important considerations in the trajectory of childhood maltreatment [3, 22].

Another finding that needs discussion here is the association of resilience with QOL. High score on resilience measure was independently associated with better QOL, even after adjusting for all other relevant factors. This is in line with most studies examining resilience and QOL in context of trauma population with lower resilience meaning lower QOL [55, 56]. One of these studies was among earthquake survivors of earthquake in China, which reported association between resilience and QOL, however, the relation was partly mediated by social support [55]. A systemic review conducted among conflict- driven adults with forced migration reported that resilience was generally related to better mental health in displaced populations with a caution that the evidence was limited [57]. Furthermore, another study from Toronto in homeless and mentally ill adults concluded that resilience and quality of life were positively correlated even after adjusting for major demographic and clinical parameters [27].

A study conducted in China among patients with bladder cancer reported that resilience was a key factor along with social support and hope, that predicted QOL, though the nature of trauma was different compared to ours [58]. Studies exploring resilience, however, need to be examined critically since resilience has differing perspectives and can be studied from a multi-system approach, especially in relation to trauma and quality of life [26].

Current depression was the only clinical diagnosis independently associated with QOL outcome. This is in line with several studies after trauma exposure exploring QOL [59]. However, the finding of current generalized anxiety disorder and PTSD not related to QOL is in contrast with results from other studies [60–62]. Also, generalized anxiety disorder is usually not diagnosed separately or investigated when it comes to trauma related disorders, instead it is categorized broadly under stress and trauma related disorders. In the ICD-10 system, the diagnosis is hierarchical, which allows to make a diagnosis of depression leaving PTSD and GAD, if depression criteria are fulfilled. Again, the fact that this was a hospital-based study with small sample size could be another explanation limiting the comparison with existing studies. There is, however, robust evidence that diagnoses of anxiety disorders have significant impairment in QOL and needs to be taken into consideration [63].

This study, however, is not without limitations and therefore, the results should be interpreted with caution. Child trauma questionnaire we used though was adapted in Nepali context, the whole chunk of sexual trauma related questions was removed saying they were not appropriate to ask in the cultural context. The possibility of recall and response bias cannot be excluded as well since this was self-reported measure as well as retrospective measure. This might have confounded the estimates. Although most of the tools we used in this study such as WHOQOL-BREF, CTQ- SF and RS were translated and adapted versions already used in Nepal, they have their own culture specific dimensions. This should also be taken into consideration in interpreting the results. This is a hospital- based study of those seeking help and with history of another trauma with cross-sectional design, limiting the casual explanation and the directionality of association. Some of the estimates could have been significant due to the small sample size. Despite these, this study attempts to explore the relation between resilience and QOL in context of trauma and childhood trauma and is possibly the first in our setting. We firmly believe that this will serve as a reference in guiding similar studies on a larger scale.

## Conclusion

Almost one third of trauma exposed patients seeking treatment at psychiatry outpatient had poor quality of life. Emotional neglect during childhood and current depression diagnosed currently were independent predictors of poor QOL, while resilience was protective. Therefore, resilience building treatment packages should be the goal. We recommend longitudinal studies with better methodology and validated tools in our context to get more clarity on this critical area.

## Supporting information

**S1 Dataset.**
(DTA)

## Author Contributions

**Conceptualization:** Saraswati Dhungana.

**Data curation:** Saraswati Dhungana.

**Formal analysis:** Saraswati Dhungana, Suraj Bahadur Thapa.

**Methodology:** Rishav Koirala.

**Project administration:** Saraswati Dhungana, Rishav Koirala.

**Resources:** Rishav Koirala, Suraj Bahadur Thapa.

**Software:** Rishav Koirala, Suraj Bahadur Thapa.

**Supervision:** Saroj Prasad Ojha, Suraj Bahadur Thapa.

**Validation:** Saroj Prasad Ojha, Suraj Bahadur Thapa.

**Visualization:** Suraj Bahadur Thapa.

**Writing – original draft:** Saraswati Dhungana.

**Writing – review & editing:** Saroj Prasad Ojha, Suraj Bahadur Thapa.

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
