## [Decision Letter · Decision Letter 0]

1 Jul 2022

PONE-D-22-16425Interplay between childhood trauma, resilience, and quality of life in patients seeking treatment at a psychiatry outpatient: a cross-sectional study from NepalPLOS ONE

Dear Dr. Dhungana,

Thank you for submitting your manuscript to PLOS ONE. After careful consideration, we feel that it has merit but does not fully meet PLOS ONE’s publication criteria as it currently stands. Therefore, we invite you to submit a revised version of the manuscript that addresses the points raised during the review process.

Dear Authors Please focus on each comment during revising the paper Thanks 

We look forward to receiving your revised manuscript.

Kind regards,

Soumitra Das

Academic Editor

PLOS ONE

Journal Requirements:

Reviewers' comments:

Reviewer's Responses to Questions

**Comments to the Author**

1. Is the manuscript technically sound, and do the data support the conclusions?

Reviewer #1: Partly

Reviewer #2: Yes

Reviewer #3: Yes

2. Has the statistical analysis been performed appropriately and rigorously? 

Reviewer #1: I Don't Know

Reviewer #2: Yes

Reviewer #3: Yes

3. Have the authors made all data underlying the findings in their manuscript fully available?

Reviewer #1: Yes

Reviewer #2: Yes

Reviewer #3: Yes

4. Is the manuscript presented in an intelligible fashion and written in standard English?

Reviewer #1: Yes

Reviewer #2: Yes

Reviewer #3: Yes

5. Review Comments to the Author

Reviewer #1: Thank you for carrying out such needed research and congratulations for the submission.

Most of the feedback have been given in the reviewed word file itself.

Some of the major issues to be addressed are:

1. Definition of QoL in the opening line itself and the abstract.

2. Inclusion criteria has been mentioned as "adults" and those with "at least one trauma one month prior to visit" while the title of the study is "Interplay between childhood trauma, resilience, and quality of life in patients seeking

treatment at a psychiatry outpatient". This needs to be reviewed.

3. The data presented and discussion on resilience in the manuscript poorly supports the mention of "resilience" in the title itself. So, data and discussion should be mentioned enough to mention resilience in the title itself.

4. "Interplay" usually refers to "bidirectional/multidirectional" interactions among the "variables" of interests while this is a cross-sectional study just assessing the "association". So, it might be better to mention "association" rather than "interplay".

5. Tables on clinical diagnoses and trauma present the diagnoses and trauma to be "pure" rather than "overlapping" while naturally, these are mostly comorbid/overlapping. So, it would be better to mention about the overlapping of diagnoses in the text/table. e.g.- depression is often comorbid with PTSD which is not reflected in the findings. This also has another major implication- the findings of association of depression only with QoL might have been because of comorbidity of depression and PTSD rather than depression alone and this needs a serious discussion to substantiate the findings.

6. It would be better to add more discussion and analysis on the "surprising" finding of association of only "emotional neglect" and "higher SES" with poor QoL. The authors need to explore the confounding variables associated with these findings- e.g.- did those with higher SES have more depression/emotional neglect/childhood trauma or other factors commonly associated with poor QoL and this findings of "higher SES" has appeared as an apparent "proxy indicator" of those underlying other factors analysed/not analysed in the study?

7. The overall English language is good but there are scopes for improvement, some have been mentioned in the attached review.

Reviewer #2: 1. very well written and conducted research relevant to Nepalese context

2. Could the authors mention how a sample size of 100 was reached and if possible please explain it in the methodology section

3. Only the D E and K sections of WMH CIDI version were chosen for psychiatric disorders. However in clinical practice and researches also suggest the presence of Dissociative Disorders in relation to childhood trauma. Could the authors mention why Dissociative Disorder was not considered?

4. There is a mention of cultural adaptation purposes as a reason to why sexual abuse of the CTQ-SF was excluded. Can the authors elaborate more to this explanation as to why sexual abuse was excluded as this removes a chunk of the population who would have been victims of sexual abuse thus not reflecting the general population of those who had experienced child trauma.

5. Finally, despite being a cross sectional study could the authors mention if any interventions were done in this population as trauma and psychiatric disorders associated with it obviously needs some kind of interventions (therapy/medications) etc..

Reviewer #3: 1. We would be interested to know why dyslexia patients were excluded.

2. How did the authors come to conclusion of 100 participants to be included. Was there any basis for sample size? Also how was the sampling done. It would be important to know. Do the authors have data on how many participants were excluded (if not not an issue).

3. Ethical approval from IRB/IRC was taken. Please mention the reference number.

4. Almost all the variables are categorized in binary. Is there any specific reason or just for the ease of statistical analysis?

5. "possibly the first study" is a big claim to make. There may be other unpublished work so this part may be removed.

6. "Trauma exposure is common globally but those seeking treatment are a high-risk group. QOL is a robust measure of health that goes beyond the traditional morbidity indicators." This part is not the conclusion of the study. May be removed or may add in the introduction section.

7. One of the most important aspect to be looked is the use of scales that have been developed from the western perspectives. The QOL, trauma, resilience all have their own culture specific dimensions and it is very difficult to adjust them via translation. Hence this part must be discussed/ acknowledged.

6. PLOS authors have the option to publish the peer review history of their article (what does this mean?). If published, this will include your full peer review and any attached files.

Reviewer #1: **Yes: **Madhur Basnet, MD(Psychiatry), Associate Professor, Dept. of Psychiatry, B. P. Koirala Institute of Health Sciences, Dharan, Nepal

Reviewer #2: No

Reviewer #3: **Yes: **Pawan Sharma

---

## [Author Response · Author response to Decision Letter 0]

26 Jul 2022

July 26, 2022

Dear Soumitra Das

Academic Editor

PLOS ONE

We thank you and the reviewers for thoroughly reviewing our manuscript and providing opportunity to revise it. The comments and questions from you and all three reviewers have been extremely valuable to improve the quality of our paper. 

We have incorporated almost all the comments from the three reviewers to the best of our knowledge. For a very few questions where we have not done so, we have provided an explanation. 

We have thoroughly revised our manuscript from title to abstract to the discussion. The references have been revised accordingly. In doing so, we have cited two more references in the discussion section and one in the methods section in sample size calculation. While reviewing our manuscript, we came across some errors and we have corrected them too in the reviewed draft.

Please find our point to point responses to the reviewers’ comments as follows with the lines and page numbers in the word tracked format of the revised manuscript, where the changes have been made, along with the excerpt of revised text where, applicable. We have also made the editor’s and reviewers’ comments in boldface and our responses in non-boldface for easy readability. 

We have submitted the revised manuscript word tracked change format, clean copy format and the rebuttal letter addressing all the comments from reviewers as a word file named “Response to reviewers.” If there are further comments/ and questions from the reviewers, we are very happy to address them.

Thank you

Saraswati Dhungana (on behalf of all co-authors)

PONE-D-22-16425

Journal Requirements:

https://journals.plos.org/plosone/s/file?id=ba62/PLOSOne_formatting_s ample_title_authors_affiliations.pdf

Authors’ response: We have rechecked the PLOS ONE’s style requirements for the manuscript and for file naming and ensure that they are in order now.

Authors’ response: Thank you for this important observation. We now have provided the correct grant numbers for the awards received for the study in the ‘Funding Information’ section such that the grant information provided in the Funding information and ‘Financial disclosure’ match as follows.

“SBT received the funding from The Norwegian Partnership Programme for Global

Academic Cooperation (NORPART). NORPART grant 2018/10039 from Norwegian Agency for International Cooperation and Quality Enhancement in Higher Education (DIKU)”.

However, the funding agency was not involved in research designing, data collection and analysis, planning and/ or manuscript preparation or decision to publish.” in lines 365-370 under funding section in the manuscript in page number 22.

We also have now included the same statement in the Financial disclosure section to match them since the previous disclosure had no grant number included. 

Authors’ response: Thank you for the comment. We now have included complete ethics statement in the “Procedure” of “Methods” section of the manuscript as follows:

“Patients fulfilling the inclusion criteria for the study and who consented to participate were provided detailed information regarding the project in verbal and in written form. They were also informed about their voluntariness in willing to withdraw from the study at any time during data collection without the need for justification.” in line 137-141 in page number 7 and 8.

“Ethical approval for this study was obtained from the Nepal Health Research Council (NHRC) (reference number 801) and Institutional Review Committee (IRC) at the Institute of Medicine (reference number 480 (611)62/ 075/076) in Nepal and Research Ethical Committee (REK) in Norway (reference number 2015/2081).” in line 144-148 in page number 8.

Response to reviewers’ comments:

Reviewer #1

Reviewer #1: Thank you for carrying out such needed research and congratulations for the submission.

Authors’ response: Thank you for the positive comment. We appreciate it.

Most of the feedback have been given in the reviewed word file itself.

Authors’ response: Thank you. We have checked the reviewed word file and made necessary corrections as suggested. We have edited all the grammatical errors you pointed out and have amended them in the revised manuscript. 

For specific comments such as Better to categorize age into developmentally appropriate categories (based on life stages) than the “mean age”, we have now categorized age as less or equal to 24 and more than 24 years in line number 153 in Procedure section in page number 8. 

Accordingly, we also modified our tables 1 and 4 at places where age categories were specified. For table 4, we reanalyzed again with this new age category and therefore, we have replaced table 4 with new one since all the values obtained had to be replaced as follows: 

 Variables (n=98)

 Bivariate analysis (Unadjusted model) Multivariate analysis (Adjusted model)

 OR (CI)

 OR (CI)

Age (<24 year)

 0.66 (0.22, 2.03) .17 (0.02, 1.25)

Gender (Male)

 1 (0.43, 2.31) 0.96 (0.25, 3.72)

SES (Lower)

 0.37 (0.16, 0.87)* 0.61 (0.16, 2.34)

CTQ Emotional neglect (No)

 0.30 (0.11, -0.87)* 0.07 (0.01, 0.43)**

Current depression (No)

 0.09 (0.03, 0.24)*** 0.05 (0.01, 0.26)***

Current PTSD (No)

 0.80 (0.26, 2.44) 2.62 (0.46, 14.80)

Current GAD (No)

 0.53 (0.23, 1.25) 0.87 (0.21, 3.55)

Mean resilience score

 3.70 (2.13, 6.44)*** 4.52 (1.97, 10.40)***

Some of the major issues to be addressed are:

1. Definition of QoL in the opening line itself and the abstract.

Authors’ response: Thank you for the comment. We now have included the definition of QOL in the opening line in introduction itself in line 88-91 in page number 5 as “Quality of life (QOL) is defined by the World Health Organization as "an individuals’ perception of their position in life in the context of the culture and value systems in which they live and in relation to their goals, expectations, standards and concerns".”

” and in the abstract in line 30-33 in page number 2 as “Quality of life is defined by the World Health Organization as " Individuals’ perception of their position in life in the context of the culture and value systems in which they live and in relation to their goals, expectations, standards and concerns".

2. Inclusion criteria has been mentioned as "adults" and those with "at least one trauma one month prior to visit" while the title of the study is "Interplay between childhood trauma, resilience, and quality of life in patients seeking treatment at a psychiatry outpatient". This needs to be reviewed.

Authors’ response: Thank you for the important observation. We would like to clarify it since this seems to have created some confusion. Our inclusion criteria were “adults” and those with “at least one trauma prior to visit” as pointed out and mentioned in the methods section in the manuscript. However, in title of the study, when we say, "Interplay between childhood trauma, resilience, and quality of life in patients seeking treatment at a psychiatry outpatient", we are referring to childhood trauma, specifically and not the trauma in general. So, trauma is one of the two necessary inclusion criteria, however, childhood trauma was not present in all patients included. History of childhood trauma was obtained by inquiring with the adults who were provided with child trauma questionnaire- short form (CTQ-SF) for categorization of the various forms of childhood trauma. In this paper, we were interested in learning about the role of childhood trauma and resilience in those adults with trauma history. Now, we hope we have made it clear. 

3. The data presented and discussion on resilience in the manuscript poorly supports the mention of "resilience" in the title itself. So, data and discussion should be mentioned enough to mention resilience in the title itself.

Authors’ response: Thank you for the thoughtful comment. We now have provided more literature on resilience in the discussion section to provide more support to the inclusion of term “resilience” in the title. We, however, agree that the title is best presented replacing the word “Interplay between” with “association of” and have done that accordingly.

We were not able to find many studies specifically examining resilience and quality of life in context of trauma and psychiatric disorders post trauma. However, we were able to find the following studies relating resilience with QOL in different populations such as displaced populations, earthquake survivors and homeless and mentally ill and we believe they are relevant to be included. We, therefore, have inserted them as appropriate in the discussion section in lines 306-314 in page number 19.

“One of these studies was among earthquake survivors of earthquake in China, which reported association between resilience and QOL, however, the relation was partly mediated by social support [55]. A systemic review conducted among conflict- driven adults with forced migration reported that resilience was generally related to better mental health in displaced populations with a caution that the evidence was limited [57]. Furthermore, another study from Toronto in homeless and mentally ill adults concluded that resilience and quality of life were positively correlated even after adjusting for major demographic and clinical parameters [58].”

4. "Interplay" usually refers to "bidirectional/multidirectional" interactions among the "variables" of interests while this is a cross-sectional study just assessing the "association". So, it might be better to mention "association" rather than "interplay".

Authors’ response: Thank you. We agree with your comment that interplay refers to “bidirectional/ multidirectional” interactions which we have not examined in this paper. We, therefore, have replaced the word “interplay” with “association” as suggested. We now have changed the title of the manuscript from “ Interplay between childhood trauma, resilience, and quality of life in patients seeking treatment at a psychiatry outpatient: a cross-sectional study from Nepal” to “Association of childhood trauma, and resilience, with quality of life in patients seeking treatment at a psychiatry outpatient: a cross-sectional study from Nepal.” in line number 1-2 in the title in page number 1.

5. Tables on clinical diagnoses and trauma present the diagnoses and trauma to be "pure" rather than "overlapping" while naturally, these are mostly comorbid/overlapping. So, it would be better to mention about the overlapping of diagnoses in the text/table. e.g.- depression is often comorbid with PTSD which is not reflected in the findings. This also has another major implication- the findings of association of depression only with QoL might have been because of comorbidity of depression and PTSD rather than depression alone and this needs a serious discussion to substantiate the findings.

Authors’ response: Thank you for the comment. We completely agree that the diagnoses can be overlapping and there were comorbid diagnoses in our study too. Of all the study participants, sixteen had current PTSD, thirty-two had current depression, while thirty-eight had current anxiety disorders. In terms of comorbidities, seven participants had comorbid PTSD and depression, whereas seven had comorbid PTSD and anxiety disorders. We did reanalysis after your comment with the comorbid diagnosis and have also added the values in table 3 at the end in page numbers 13 and 14, with addition of two rows as follows. 

However, there was not statistically significant finding in bivariate analysis. So, we did not include them in further analysis in association with QOL and therefore, we have not modified other tables.

PTSD+depression 

No

Yes 

29

4 

60

6 

0.22 

0.19, 2.77

PTSD+GAD

No

Yes 

31

2 

61

5 0.08 0.23, 6.92

6. It would be better to add more discussion and analysis on the "surprising" finding of association of only "emotional neglect" and "higher SES" with poor QoL. The authors need to explore the confounding variables associated with these findings- e.g.- did those with higher SES have more depression/emotional neglect/childhood trauma or other factors commonly associated with poor QoL and this findings of "higher SES" has appeared as an apparent "proxy indicator" of those underlying other factors analysed/not analysed in the study?

Authors’ response: Thank you for the important observation. We now have clarified our statements "surprising" finding of association of only "emotional neglect" and "higher SES" with poor QoL as follows.

We have inserted the following text in the discussion in line 275-278 in page number 17. 

“Childhood trauma is related to poor QOL regardless of the type of trauma. This was also one of our hypotheses on starting to write the paper, but during analysis, we found that only emotional neglect was associated with QOL, unlike in most studies.” We, therefore, used the term surprising finding. 

In regard to higher SES associated with poor QOL, we have discussed this further in the manuscript in line 262-266 in page 17 under discussion section as follows

“Exploring other confounders might have given a more accurate picture. The possibility of having more depression/ emotional neglect/ childhood trauma or other relevant factors such as social support, in those with higher SES is also a valid concern prone to give biased estimates in terms of association with QOL.”

7. The overall English language is good but there are scopes for improvement, some have been mentioned in the attached review.

Authors’ response: Thank you for the comment. We now have made necessary edits as suggested in the attached review at all places.

Reviewer #2

Reviewer #2: 1. very well written and conducted research relevant to Nepalese context

Authors’ response: Thank you for the positive comment. 

2. Could the authors mention how a sample size of 100 was reached and if possible, please explain it in the methodology section

Authors’ response: Thank you for the important observation. We now have explained how we came up with the sample size of 100 as follows and have included this in the methods section in line 124- 131 in page 7 as follows. 

“We used purposive sampling technique for the study participants. Sample size was calculated based on the Cochran’s formula of sample size calculation for cross- sectional studies as follows: 

n=(z1−α/2)2∗(p)(q)/(d)2,n=(z1−α/2)2∗(p)(q)/(d)2. Here, p= 50% from previous study [30], (z1− α /2)2 = 1.96 for 95% confidence interval, and d= 10%, precision of estimate. With this, the sample size calculated, n= 96.04 ~ 96. Considering 10% dropouts, we calculated the total required sample size as 96+10= 106. We, however, were able to recruit only 100 patients in our study.”

We also deleted the text “The total number of participants included in the study was one hundred.” in line 123-124. 

3. Only the D E and K sections of WMH CIDI version were chosen for psychiatric disorders. However in clinical practice and researches also suggest the presence of Dissociative Disorders in relation to childhood trauma. Could the authors mention why Dissociative Disorder was not considered?

Authors’ response: Thank you for the comment. We agree that dissociative disorders are common in relation to childhood trauma, and it would have been interesting to see how they develop in these patients. However, since in this particular paper, our focus was mainly on three clinical diagnoses as depression, anxiety disorder and post-traumatic stress disorder, we therefore did not consider Dissociative Disorders.

4. There is a mention of cultural adaptation purposes as a reason to why sexual abuse of the CTQ-SF was excluded. Can the authors elaborate more to this explanation as to why sexual abuse was excluded as this removes a chunk of the population who would have been victims of sexual abuse thus not reflecting the general population of those who had experienced child trauma.

Authors’ response: Thank you for the comment. Yes, we have mentioned that sexual abuse of the CTQ-SF was excluded during cultural adaptation purposes, and we also agree that this limits generalization of the findings to general population with those with child trauma experience. 

We now have elaborated more to why this was excluded by inserting the following “In Nepali culture, questions regarding sexual abuse are considered inappropriate and offensive even to this date, making the participants uncomfortable.” in line 176-178 in page number 9.

5. Finally, despite being a cross sectional study could the authors mention if any interventions were done in this population as trauma and psychiatric disorders associated with it obviously needs some kind of interventions (therapy/medications) etc..

Authors’ response: Thank you for the important point raised. For this population with diagnosed psychiatric disorders, since they were recruited from psychiatry outpatient, they were provided standard treatment (psychological and pharmacotherapy) as appropriate. For those requiring intensive psychotherapy, they were further referred to clinical psychologist and followed up in outpatient and treated as usual patients.

Reviewer #3

Reviewer #3: 1. We would be interested to know why dyslexia patients were excluded.

Authors’ response: Thank you for the comment. Dyslexia is a learning disorder that involves difficulty reading due to problems identifying speech sounds and learning how they relate to letters and words (decoding). In our study, there were several questionnaires that needed reading on part of the participants and because of the condition, this might pose greater problems on carrying out interviews. We, therefore, excluded dyslexia patients from the study. 

2. How did the authors come to conclusion of 100 participants to be included. Was there any basis for sample size? Also how was the sampling done. It would be important to know. Do the authors have data on how many participants were excluded (if not not an issue).

Authors’ response: Thank you for the comment. We now have included in the methods section how we came up with the sample size of 100 and the sampling design. We have inserted the following statement in line 124-131 in page 7 under Participants in Methods section.

“We used purposive sampling technique for the study participants. Sample size was calculated based on the Cochran’s formula of sample size calculation for cross- sectional studies as follows: 

n=(z1−α/2)2∗(p)(q)/(d)2,n=(z1−α/2)2∗(p)(q)/(d)2. Here, p= 50% from previous study [34], (z1− α /2)2 = 1.96 for 95% confidence interval, and d= 10%, precision of estimate. With this, the sample size calculated, n= 96.04 ~ 96. Considering 10% dropouts, we calculated the total required sample size as 96+10= 106. We, however, were able to recruit only 100 patients in our study”. 

We also deleted the text “The total number of participants included in the study was one hundred.” in line 123-124. 

3. Ethical approval from IRB/IRC was taken. Please mention the reference number.

Authors’ response: Thank you for the comment. We now have included the reference number of all ethical approvals obtained as follows in line 144-148 in page 8 under Procedure of Methods section.

“Ethical approval for this study was obtained from the Nepal Health Research Council (NHRC) (reference number 801) and Institutional Review Committee (IRC) at the Institute of Medicine (reference number 480 (611)62/ 075/076) in Nepal and Research Ethical Committee (REK) in Norway (reference number 2015/2081).”

4. Almost all the variables are categorized in binary. Is there any specific reason or just for the ease of statistical analysis?

Authors’ response: Thank you for the comment. Yes, almost all the variables are categorized as binary. There is no specific reason for this categorization.

5. "possibly the first study" is a big claim to make. There may be other unpublished work so this part may be removed.

Authors’ response: Thank you for the comment. We agree "possibly the first study" is a big claim to make when there are so many unpublished studies. We, therefore, have removed this part as suggested in line 248 in page 16 under the section on discussion.

6. "Trauma exposure is common globally but those seeking treatment are a high-risk group. QOL is a robust measure of health that goes beyond the traditional morbidity indicators." This part is not the conclusion of the study. May be removed or may add in the introduction section.

Authors’ response: Thank you for the important observation. "Trauma exposure is common globally but those seeking treatment are a high-risk group. QOL is a robust measure of health that goes beyond the traditional morbidity indicators." This part as you suggested might not fit in the conclusion of the study, so we removed it from the conclusion section from line 355- 357 in page 21.

7. One of the most important aspect to be looked is the use of scales that have been developed from the western perspectives. The QOL, trauma, resilience all have their own culture specific dimensions and it is very difficult to adjust them via translation. Hence this part must be discussed/ acknowledged.

Authors’ response: Thank you for the comment. We agree that the use of the scales that have been developed from the western perspectives might be difficult to adjust via translation alone to be used locally. The WHOQOL- BREF scale used in this study has been cross culturally validated in multiple settings, including low and middle income countries as much as high income countries and has undergone rigorous methodology. Similarly, other scales such as resilience and Child trauma questionnaire- short form (CTQ-SF) for child trauma assessment have been validated for use in Nepal by scholars with years of experience working with translation and adaptation with several collaborative projects in Nepal. However, we strongly agree with your comment that all the scales have their own culture specific dimensions and therefore, we have acknowledged this in the discussion section in line 340-344 in page 20 and 21 as follows. 

“Although most of the tools we used in this study such as WHOQOL-BREF, CTQ- SF and RS were translated and adapted versions already used in Nepal, they have their own culture specific dimensions. This should also be taken into consideration in interpreting the results.”

---

## [Decision Letter · Decision Letter 1]

20 Sep 2022

Association of childhood trauma, and resilience, with quality of life in patients seeking treatment at a psychiatry outpatient: a cross-sectional study from Nepal

PONE-D-22-16425R1

Dear Dr. Dhungana,

We’re pleased to inform you that your manuscript has been judged scientifically suitable for publication and will be formally accepted for publication once it meets all outstanding technical requirements.

Kind regards,

Soumitra Das

Academic Editor

PLOS ONE

Additional Editor Comments (optional):

Reviewers' comments:

Reviewer's Responses to Questions

**Comments to the Author**

1. If the authors have adequately addressed your comments raised in a previous round of review and you feel that this manuscript is now acceptable for publication, you may indicate that here to bypass the “Comments to the Author” section, enter your conflict of interest statement in the “Confidential to Editor” section, and submit your "Accept" recommendation.

Reviewer #2: All comments have been addressed

Reviewer #3: All comments have been addressed

2. Is the manuscript technically sound, and do the data support the conclusions?

Reviewer #2: (No Response)

Reviewer #3: Yes

3. Has the statistical analysis been performed appropriately and rigorously? 

Reviewer #2: (No Response)

Reviewer #3: Yes

4. Have the authors made all data underlying the findings in their manuscript fully available?

Reviewer #2: (No Response)

Reviewer #3: Yes

5. Is the manuscript presented in an intelligible fashion and written in standard English?

Reviewer #2: (No Response)

Reviewer #3: Yes

6. Review Comments to the Author

Reviewer #2: (No Response)

Reviewer #3: (No Response)

7. PLOS authors have the option to publish the peer review history of their article (what does this mean?). If published, this will include your full peer review and any attached files.

Reviewer #2: **Yes: **Utkarsh Karki

Reviewer #3: **Yes: **Dr. Pawan Sharma

---

## [Editor Report · Acceptance letter]

23 Sep 2022

PONE-D-22-16425R1 

*Association of childhood trauma, and resilience, with quality of life in patients seeking treatment at a psychiatry outpatient: a cross-sectional study from Nepal*

Dear Dr. Dhungana:

I'm pleased to inform you that your manuscript has been deemed suitable for publication in PLOS ONE. Congratulations! Your manuscript is now with our production department. 

Kind regards, 

on behalf of

Dr. Soumitra Das 

Academic Editor

PLOS ONE